# Physical Activity and Nutrition: Two Promising Strategies for Telomere Maintenance?

**DOI:** 10.3390/nu10121942

**Published:** 2018-12-07

**Authors:** Estelle Balan, Anabelle Decottignies, Louise Deldicque

**Affiliations:** 1Institute of Neuroscience, Université catholique de Louvain, Place Pierre de Coubertin 1 L8.10.01, 1348 Louvain-La-Neuve, Belgium; estelle.balan@uclouvain.be; 2De Duve Institute, Université catholique de Louvain, Avenue Hippocrate 75, 1200 Brussels, Belgium; anabelle.decottignies@uclouvain.be

**Keywords:** aging, exercise, diet, telomerase, TERRA, telomere length, senescence

## Abstract

As the world demographic structure is getting older, highlighting strategies to counteract age-related diseases is a major public health concern. Telomeres are nucleoprotein structures that serve as guardians of genome stability by ensuring protection against both cell death and senescence. A hallmark of biological aging, telomere health is determined throughout the lifespan by a combination of both genetic and non-genetic influences. This review summarizes data from recently published studies looking at the effect of lifestyle variables such as nutrition and physical activity on telomere dynamics.

## 1. Introduction

The proportion of the world population aged 60 years and over is increasing rapidly and is projected to rise above 20% in 2050, which will exceed the number of children in the world [1,2]. Indeed, most countries are seeing their demographic structure getting older. The aging of the population has major implications socially and economically as aging is characterized by a progressive loss of physiological integrity, leading to impaired function and autonomy [3]. This functional decline is the greatest risk factor for conditions that limit health span, i.e., quality of life at old age, and for the majority of chronic diseases such as type 2 diabetes, Alzheimer’s disease, and various cancers [4,5]. Notably, senescence has become the greatest risk factor for death in developed countries [6]. With the increasing longevity, the maintenance of health and autonomy at old age becomes crucial. Today more than ever, highlighting strategies to counteract age-related disorders is a major public health concern.

Geroscience is an area that aims to explain the biological mechanisms of aging. Aging research has experienced an unprecedented advance over recent years, particularly with the discovery that the rate of aging is controlled, at least to some extent, by genetic pathways and biochemical processes conserved in evolution such as genomic instability, telomere attrition, epigenetic alterations, loss of proteostasis, deregulated nutrient sensing, mitochondrial dysfunction, cellular senescence, stem cell exhaustion, and altered intercellular communication [3]. Recent findings have revealed the importance of the regulation of telomere length and integrity during the aging process [7], as well as potential interventions to improve the health span such as physical activity and healthy diet [8]. Telomere attrition is associated with decreased life expectancy and increased risk of chronic disease [9] and has been described as one of the most important biological hallmarks of aging due to a key role in cellular senescence [3]. During the past decade, telomeres have evolved from a simple capsule hiding the ends of chromosomes to complex nucleoprotein structures with an active role in the protection of the genome and in the regulation of cellular senescence [10,11]. While previous reviews specifically focused on the regulation of telomere length by either nutrition [12] or exercise [13,14], the present review will give a broader view looking at the impact of lifestyle variables on human telomere dynamics with emphasis on diet and physical activity.

## 2. Telomeres

Mammalian telomeres consist of repetitive DNA G- and C-rich sequences (5′-TTAGGG-3′/3′-CCCTAA-5′) with the 3′ end of the G-strand extending beyond the 5′ end [15]. The double-stranded telomeric DNA is bound by the six-subunit shelterin complex: telomeric repeat factor 1 (TRF1), telomere repeat factor 2 (TRF2) and protection of telomere 1 (POT1) directly recognize TTAGGG repeats and they are interconnected with TRF1- and TRF2-interacting nuclear protein 2 (TIN2), POT1 and TIN2-interacting protein (TPP1) and repressor/activator protein 1 (RAP1) [16]. The Shelterin complex facilitates the formation of a lariat-like structure with a T- and a D-loop, allowing the telomere end to be hidden (Figure 1). This conformation represses the DNA damage response (DDR) at telomeres, thereby preventing the activation of the ataxia telangectasia mutated (ATM) and RAD3-related (ATR) kinases that induce cell cycle arrest in response to DNA double-strand breaks and other types of DNA damage [10,17]. 

Many types of human cells lack telomerase, the enzyme responsible for telomere synthesis by adding nucleotides to the chromosome ends [18]. Telomerase consists of two core components; the catalytic subunit, telomerase reverse transcriptase (TERT) [19] and a RNA template (TERC) [20]. Hence, because of the “end-replication problem”, i.e., DNA polymerase incapacity to maintain telomere length during cell divisions, somatic cells display gradual telomere shortening with age [21]. Critical loss of telomeric DNA or unprotected telomeres leads to insufficient chromosomes end protection and to the activation of the DNA damage response [17]. Damage at telomeres can also happen independently of cell division, notably in response to the accumulation of oxidative lesions, smoking behavior, or obesity [22]. While telomere shortening is considered as a protection against tumor development, loss of telomere function induces cellular senescence and impairs tissue turnover leading to the aging of the whole organism. The process of telomere attrition is not constant and differs between people [23,24], which can be explained by the impact of inflammation and oxidative stress on telomere shortening, which differs from one individual to the other [9]. Globally, telomere health is determined throughout the lifespan by a combination of both genetic and non-genetic factors.

## 3. Telomere Regulation by Nutrition

Lifestyle factors such as an unhealthy diet, physical inactivity, or smoking habits have been related to shorter leukocyte telomere length, a biomarker of the “biological age” of cells, as opposed to the “chronological age” [25]. Some studies have reported an association between diet [26,27,28,29,30] or consumption of specific foods [31] and leukocyte telomere length. To note, the rates of telomere shortening are similar in leukocytes and somatic cells, so that telomere length in leukocytes is now accepted to be representative of global telomere length in somatic cells [32].

### 3.1. Consumption of Specific Foods

Telomere length is positively associated with the consumption of legumes, nuts, seaweed, fruits, and 100% fruit juice, dairy products, and coffee, whereas it is inversely associated with consumption of alcohol, red meat, or processed meat [27,28,33,34]. Telomere attrition may represent a mechanism by which large sugar intake accelerates cardiometabolic disease [35]. Several studies suggest that reducing sugary beverage consumption could be associated with extended telomere length, independently of other characteristics such as age, sex, or body mass index [26,27,28]. Those results indicate that leukocyte telomere length maintenance may be sensitive to the metabolic effects of high sugar consumption over time [26]. Leung et al. examined the associations between the consumption of sugar-sweetened beverages (including soda, soft drinks, fruit-flavored drinks, sports drinks, and energy drinks), diet soda, fruit juice, and leukocyte telomere length in 5309 adults aged 20–65 years from the United States without any history of diabetes or cardiovascular disease [28]. After adjustment for sociodemographic and health-related characteristics, the consumption of sugar-sweetened beverages was associated with shorter telomeres, whereas the consumption of 100% fruit juice was associated with a higher telomere length. No significant association was observed between consumption of diet soda and telomere length [28]. As cross-sectional study may not be the most appropriate study design to assess telomere length, more recently, the same group conducted a longitudinal study to evaluate the associations between sugary foods and beverages and leukocyte telomere length in 65 overweight and obese pregnant women aged between 18 and 45 years. From ≤16 weeks gestation to 9 months postpartum, dietary intake was monitored using 24-h diet recalls and leukocyte telomere length was measured by real-time quantitative polymerase chain reaction (qPCR). From the baseline to 9 months post-partum, a low consumption of sugar-sweetened beverages was associated with longer leukocyte telomere length but no association was found between sugary foods and leukocyte telomere length [26].

People who regularly eat beans and whole grains are frequently spotlighted for increased longevity [36]. Boressen et al. (2016) tried to determine the feasibility of increasing navy beans or rice bran intake in colorectal cancer survivors to increase dietary fiber. The authors hypothesized that an increased amount of dietary fiber could positively regulate telomere length. Twenty-nine volunteers participated to a randomized-controlled trial with foods that included cooked navy beans powder (35 g/day), heat-stabilized rice bran (30 g/day), or no additional ingredient. The amount of navy beans powder or heat-stabilized rice bran consumed represented 4–9% of daily caloric intake. Over the intervention period of 4 weeks, no major gastrointestinal issues were reported and the dietary fiber amounts increased in the navy beans and rice bran groups at weeks 2 and 4 compared to baseline and the control group. At baseline, peripheral blood mononuclear cell (PBMC) telomere length was positively correlated with high density lipoprotein (HDL)-cholesterol and negatively correlated with lipopolysaccharide and age. Although a higher consumption of navy beans (35 g/day) or rice bran (30 g/day), known to contain fiber, iron, zinc, thiamin, niacin, vitamin B6, folate, and alpha-tocopherol, did not influence PBMC telomere length after the short intervention period of 4 weeks [31], the effect of a fiber-enriched diet on telomere length should be investigated in a healthy population over a longer period of time. This may be highly relevant in the context of colorectal cancer known to be associated with dysfunctional telomeres [37].

### 3.2. Diet Composition

While it is important to be aware of the effects of individual foods, it is even more critical to assess the role of cumulative nutrients contained in specific diets on telomere length, which better reflects reality. In 2015, Lee et al. compared the influence of the dietary pattern on leukocyte telomere length [27]. Dietary data were collected from a semi-quantitative food frequency questionnaire at baseline and leukocyte telomere length was assessed using qPCR 10 years later. A total of 1958 middle-aged and older Korean adults (40–69 years at baseline) were included in the study. The authors identified two major dietary patterns: “the prudent dietary pattern” was characterized by a high intake of whole grains, fish and seafood, legumes, vegetables, and seaweed, whereas the “western dietary pattern” included a high intake of refined grain, red meat or processed meat, and sweetened carbonated beverages. Using a multiple linear regression model adjusted for age, sex, body mass index, and other potential confounding variables, the “prudent dietary pattern” was found to be positively associated with leukocyte telomere length while an inverse trend was found in the “western dietary pattern”. These results suggest that diet in the remote past, that is, 10 years earlier, may affect the degree of biological aging in middle-aged and older adults [27]. 

One of the best models of healthy eating is the Mediterranean diet which is characterized by a high intake of vegetables, legumes, nuts, fruits, and cereals (mainly unrefined); a moderate to high intake of fish; a low intake of saturated lipids but high intake of unsaturated lipids, particularly olive oil; a regular but moderate intake of alcohol, specifically wine [38]. This diet has been shown to prevent age-associated telomere shortening [29,30,39] and has been associated with reduced mortality risk in older people [40]. The possible mechanisms for the protective effect of the Mediterranean diet on telomeres will be discussed in the next section. In 4676 healthy women (42–70 years), the higher scores on the Mediterranean diet, evaluated by food frequency questionnaires, were associated with longer leukocyte telomere length [30]. In the same study, no association between prudent or western dietary patterns and telomere length was observed [30], while a prudent diet was previously found to be positively and a western diet negatively associated with leukocyte telomere length in 1958 middle-aged and older women and men [27]. Similarly, in 217 men and women aged 71–87 years, a greater adherence to a Mediterranean diet was associated with longer leukocyte telomere length and higher PBMC telomerase activity [29]. However, a recent study in 679 Australian men and women (57–68 years) found no association between diet quality and whole blood telomere length, including the Mediterranean diet. In this study, the authors assessed the dietary intake by using a 111-item food frequency questionnaire, which assessed self-reported intake of foods and beverages over the last 6 months, and the diet quality by three indices: the Dietary Guideline Index (DGI), the Recommended Food Score (RFS), and the Mediterranean Diet Score (MDS) [41]. Whole blood telomere length did not differ by age, smoking status, BMI, or physical activity but women had longer telomeres than men [41]. The discrepant results between studies could be explained by the use of different questionnaires to assess the diet quality and/or the populations studied. Longitudinal studies may be more suitable to determine the potential positive influence of diet on telomere health. 

Of note, in animal models, calorie restriction has been shown to have a positive effect on telomere length [42] and to globally delay the onset of aging and age-related disease such as diabetes, cardiovascular diseases, various neurological disorders, cancer, and obesity [43,44], possibly via a reduction in oxidative stress [45,46]. In humans, the data are less convincing, probably because decreasing the caloric intake by a third or a half is very challenging in that population, certainly in the long-term.

Having presented which foods and diets were potentially beneficial for telomere health in general, the next section will attempt to summarize the mechanisms involved in those effects.

### 3.3. Mechanisms

Unhealthy dietary habits have been linked to an inflammatory state, contributing to progressive telomere attrition [47]. As unhealthy dietary habits increase the production of reactive oxygen species (ROS), it is possible that the impact on telomere erosion goes through an increased oxidation of telomeric DNA. Supporting this, is the observation that, because of their high content in guanine residues, telomeric sequences are highly prone to oxidation into 8-oxoG, at least in in vitro experiments [48]. When present at telomeres, 8-oxoG residues are likely decreasing the affinity of shelterin proteins for telomeric DNA and are, as well, disrupting the G-quadruplex structures of telomeres that play important roles at telomeres, like the regulation of telomerase activity [49]. Altogether, it is therefore possible that nutrients regulate telomere health by regulating oxidative stress and systemic inflammation [50]. Globally, it can be hypothesized that any antioxidant or anti-inflammatory diet could be protective for telomeres by slowing down telomeric shortening and delay the aging process. The intake of nutrients having antioxidant and anti-inflammatory properties, such as vitamin C or E, polyphenols, curcumin, or omega-3 fatty acid, has been associated with longer telomeres, at least in mouse [51]. 

The positive effects of the Mediterranean diet on telomeres may be due to its antioxidant and anti-inflammatory potential [52,53]. To understand whether the Mediterranean diet could prevent endothelial cellular senescence by regulating oxidative stress, the serum of 20 elderly subjects (age > 65 years; 10 men and 10 women) was collected before and after having randomly followed each of the 3 following diets for 4 weeks: a Mediterranean diet, a saturated fatty acid diet and a low fat and high carbohydrate diet [54]. Human endothelial cells incubated with the serum collected after ingestion of the Mediterranean diet produced lower intracellular ROS, unavoidable byproducts of aerobic metabolism, and the percentage of cells with telomere shortening was lower compared to baseline and the two other intervention diets. The authors postulated that those findings were possibly due to nutrients with antioxidant capacities included in the Mediterranean diet [54]. In 2015, a direct association was found between the pro-inflammatory capacity of the diet and telomere shortening in a population at high risk of cardiovascular disease. The diets with the higher pro-inflammatory scores were associated with a higher risk of having shorter telomeres and a two-fold risk of accelerated telomere shortening after a five-year follow-up period [47]. At a molecular level, exposure of human leukemic cells to the pro-inflammatory factor tumor necrosis factor alpha (TNFα) induced a senescence state, which was featured by prolonged growth arrest, increased beta-galactosidase activity, cyclin-dependent kinase inhibitor 1 (p21) activation, decreased telomerase activity, telomeric disturbances such as shortening, losses, and fusions, as well as additional chromosomal aberrations [55]. Those results indicate that TNFα alters telomere maintenance. Moreover, subjects with higher adherence to Mediterranean diet had lower plasmatic level of C-reactive protein (CRP), interleukin 6 (IL-6), TNFα, and nitrotyrosine, all markers of inflammation and/or oxidative stress [29]. As high levels of oxidative stress [56] and inflammation [57] are known to increase telomere attrition rate, the Mediterranean diet may protect telomere maintenance by downregulating both processes. 

While a healthy diet may have an overall positive influence on telomeres, it seems that the benefit may be reduced in some individuals with specific genetic background [58]. For example, the *rs1800629* polymorphism at the *TNFα* gene has been shown to interact with the Mediterranean diet to modify triglyceride metabolism and inflammation status in patients suffering from the metabolic syndrome [58]. At baseline, the patients with the GG alleles had higher fasting and postprandial triglyceride and higher sensitivity C-reactive protein plasma levels than the patients with the GA or AA alleles. However, those differences between the polymorphisms observed at baseline disappeared after having followed a Mediterranean diet for 12 months, suggesting that the GG carriers were highly sensitive to this specific diet. Globally, understanding the role of gene–diet interactions may be an efficient strategy for personalized treatment of specific pathologies such as metabolic syndrome.

While some molecular mechanisms have already been highlighted, further research is needed to better understand how different diets and specific foods regulate biological aging in order to develop efficient nutritional strategies according to specific populations.

## 4. Telomere Regulation by Physical Activity

This section will deliberately present a positive view regarding the effects of physical activity on telomere dynamics, but it should be kept in mind that about half of the studies dealing with that topic found no association between physical activity and telomere length [13]. Obviously, further investigation will be needed to determine why the different findings are such discrepant from one study to the other. In addition, new analytical tools need to be developed to measure telomere length more accurately as well as new biomarkers for assessing biological aging [13]. 

### 4.1. Dose-Response

The beneficial effect of physical activity on telomere length has been reviewed and discussed by Denham et al. [59]. However, there is currently no clear consensus on the optimal exercise dose to exert the most beneficial response on telomere health. The effect of 9 different modes of physical activity, and thereby intensity levels, on leukocytes telomere length has been tested in US adults (20–84 years, *N* = 6503) [60]. The only mode of physical activity displaying an association with leukocyte telomere length was running, the most intense mode in that study. Another study used the data of a subgroup of the previously mentioned cohort (*N* = 5883) and found a strong positive association between the weekly amount of physical activity and telomere length in leukocytes [61]. However, a recent study indicated that moderate amounts of exercise are sufficient to protect telomere health, while higher amounts may not elicit additional benefits [62]. In 2010, telomere length was measured in skeletal muscle of 18 experienced middle-aged endurance runners versus 19 sedentary subjects [63]. No difference between groups was found. However, telomere length in the muscle of endurance athletes was inversely related to the number of years they spent running and the hours of spent training, which indicated that high level of chronic endurance could accelerate telomere attrition and thereby biological aging. More recently, leukocyte telomere length was determined in 61 young elite athletes and 64 healthy inactive controls [64]. Even with their high intensity and training volume, the young elite athlete had longer telomeres than their inactive peers. Finally, leukocyte telomere length was 11% higher in ultra-marathon runners compared to 56 healthy subjects, matched for age [65]. Altogether, these results suggest that high amounts of exercise may not reverse the beneficial impact of exercise on telomere length but further investigation is needed to see whether tissue-specific differences exist. 

In humans, Diman et al. showed that a high intensity cycling exercise (75% VO_2_ peak) boosted the transcription of skeletal muscle telomeres more than a moderate intensity exercise (50% VO_2_ peak) of the same duration [66]. More details on the molecular mechanisms of this observation will be reported in a following section. In conclusion, due to the paucity of data, it remains unclear which of the intensity or the volume of each training session or the combination of both is crucial to induce the beneficial effects exercise has on telomere maintenance.

### 4.2. Physical Activity and Telomerase Activity

While physical activity has been associated with longer telomere length and protection against age-related telomere attrition [65,67,68,69,70,71,72], the mechanisms by which physical activity exerts its positive effects on telomeres are still largely unknown. As TERT, the catalytic subunit of the telomerase complex, is considered as the limiting factor for telomerase activity in human somatic cells, an increase in telomerase activity after exercise could promote telomere elongation. Chilton et al. were the first to look at the regulation of telomerase after one acute bout of exercise [73]. To that end, they investigated the acute exercise-induced response on telomeric-associated genes and microRNAs (miRNAs), i.e., small noncoding RNA molecules functioning in RNA silencing and post-transcriptionally regulating gene expression by base pairing with messenger RNA (mRNA). Blood samples were taken in 22 healthy young males before, immediately after, and 60 min after a 30-min bout of treadmill running at 80% VO_2_ peak. In white blood cells, both TERT and sirtuin-6 (SIRT6) mRNA levels were increased immediately after exercise. Sixty minutes post-exercise, there was an upregulation of miR-186 and miR-96 expression, two miRNA controlling the expression of genes involved in telomere homeostasis [73]. In addition, telomeric repeat binding factor 2, interacting protein (TERF2IP) was identified as a potential binding target for miR-186 and miR-96 and demonstrated concomitant downregulation with the upregulation of those 2 miRNA at 60 min post-ex. TERF2IP is part of the shelterin complex and is recruited to telomeres via interaction with TRF2 [74]. TERF2IP deletion reduces telomere stability and increases telomere recombination [75]. However, TERF2IP/RAP1 has been found to be both a negative [76] and a positive regulator of telomere length [74]. Interestingly, TERF2IP/RAP1 is also known to play additional telomere-unrelated functions through the binding to extra-telomeric sites in the genome. Several regulatory functions have been attributed to the binding of TERF2IP/RAP1 outside telomeres, including the modulation of the nuclear factor-kappa B (NF-kB)-dependent pathway [77]. Whether the non-telomeric functions of TERF2IP/RAP1 play any role after exercise however warrants further investigation.

A very recent study tested whether an acute bout of exercise would induce a different response on telomerase activity in older vs. young individuals and whether this response would be gender-specific [78]. To test this hypothesis, age- and gender-related differences in telomerase and shelterin responses at 30, 60, and 90 min after a high intensity interval cycling exercise were determined in PBMC of 11 young (22 years) and 8 older (60 years) men and women. A larger increase in telomerase activity, as assessed by TERT mRNA levels, was found in the young compared to the older group after exercise [78]. The second main finding of that study was the higher TERT response to the acute endurance exercise in men compared to women, in whom the response was negligible, independently of age. Those results showed that aging is associated with reduced telomerase activation in response to high-intensity cycling exercise in men [78]. Another study showed that a 30-min treadmill running session was long enough to increase PBMC telomerase activity in 22 young healthy subjects including 11 women and 11 men [79]. Altogether, those recent studies confirm that the increasing telomerase activity after a single bout of exercise could be one of the mechanisms by which physical activity protects against aging [73,78,79]. 

Nevertheless, the increase in telomerase activity seems transient after acute exercise. The effect of a whole training program on telomerase activity and telomere length was investigated in 68 female and male caregivers, a population known to cope with chronic high stress, physical inactivity, and dealing with a high risk of disease [80]. Half of the subjects followed an endurance training program consisting in 40 min of aerobic exercise 3–5 times per week, while the other half remained inactive for 24 weeks. In aerobic trained caregivers, the leukocyte telomere length was lengthened after training while the telomere length was slightly shortened in the inactive group, as would be expected over a six month-period. However, no change in PBMC telomerase activity after the intervention was observed in either group [80]. Together with the findings from the acute exercise studies, it can be hypothesized that exercise-induced higher telomerase activity in PBMC may be a transient mechanism returning to basal level several hours after a single bout of exercise, though the exact kinetics still needs to be determined. In addition, telomerase is not active in all cell types, which implies that other mechanisms contribute to the exercise-induced beneficial effects on telomere length and integrity in those cells.

### 4.3. Physical Activity and Oxidative Stress

It is well established that moderate and regular physical activity is able to reduce the effect of aging by alleviating oxidative stress level [81]. Recently, an inverse relationship between the aerobic capacity and oxidative stress biomarkers in the blood was found in older Mexican adults [82]. Moreover, several studies indicate that oxidative stress accelerates telomere attrition [83,84,85]. 

Mechanistically, exercise transiently upregulates ROS production, which is counteracted by an antioxidant exercise-induced systemic adaptation response to protect the cells against oxidative damage [86,87]. This antioxidant response can be explained by the hormesis concept, namely that low levels of stress stimulate existing cellular and molecular pathways that improve the capacity of cells and organisms to withstand subsequent greater stress [88]. The antioxidant response leads to the activation of redox-sensitive transcription factors such as NF-*k*B, activator-protein 1 (AP-1) [89], and co-factors such as peroxisome proliferator-activated receptor gamma coactivator 1-alpha (PGC-1α) [81,90]. As a metabolic energy deprivation sensor, AMP-activated protein kinase (AMPK) is activated by exercise and triggers PGC-1α transcription and activation by allowing its nuclear translocation [91]. Once in the nucleus, PGC-1α induces the transcription of nuclear respiratory factor 1 (NRF1), an antioxidant factor. By activating the PGC-1α redox signaling pathway, exercise stimulates mitochondrial biogenesis and ameliorates the age-related decline in mitochondrial oxidative capacity [90]. 

### 4.4. Physical Activity and Regulation of TERRA

Mature muscle cells are one example of cells in which telomerase is not active and despite the absence of telomerase activity, physical activity has been shown to influence positively telomere length in skeletal muscle [92]. In the search of additional mechanisms, telomeric repeat containing RNAs, dubbed TERRA, have emerged as particularly interesting targets. For a long time, telomeres have been considered transcriptionally silent. Yet it turns out that telomeres are transcribed into TERRA molecules [93]. Located in the nucleus, TERRA are non-coding RNAs whose transcription is initiated from subtelomeric promoters. They consist of subtelomeric-derived sequences and G-rich telomeric repeats [93,94]. Once transcribed, TERRA remain partly associated with telomeres to play crucial functions, including telomere protection [95]. Diman et al. identified NFR1 as an important regulator of human telomere transcription in cultured cells. In addition to NRF1, PGC-1α as well as AMPK were found to be important molecular intermediates in the transcription of telomeres. As AMPK can be activated by high-intensity or long-lasting endurance exercise, it was tested in vivo whether an acute endurance exercise bout could upregulate telomere transcription in human skeletal muscles. Ten healthy young volunteers were submitted to a cycling endurance exercise of either low or high intensity and three muscle biopsies were taken before, directly after, and 2 h 30 min after exercise. Phosphorylation of acetyl-Coa carboxylase (ACC), a bona fide marker of AMPK activation, was induced after exercise, especially in the high intensity group. The same pattern of activation was found for the translocation of PGC-1α to the nucleus and for TERRA induction. As telomere transcription is activated by NRF1, an antioxidant factor, the upregulation of TERRA may be part of the antioxidant response that skeletal muscles set up to counteract exercise-induced ROS production [86]. Moreover, as they consist of a high content in guanine residues prone to oxidation, TERRA may possibly shield TTAGGG telomeric repeats from ROS [66]. Together, those results suggest that an acute bout of endurance exercise is sufficient to induce telomere transcription that, on a longer term, could possibly provide a mechanism for TERRA renewal and telomere protection in skeletal muscle.

## 5. Conclusions

Nowadays, the aging of the world population has major social and economic implications. Today more than ever, highlighting strategies to counteract age-related diseases is a major public health concern. In this review, we explored data from recently published studies looking at the influence of lifestyle variables such as nutrition and physical activity on one of the most important hallmarks of aging: the telomere. 

Most studies indicate an important role of diet on the degree of biological aging. Indeed, a healthy diet characterized by a high intake of dietary fiber and unsaturated lipids exerts a protective role on telomere health, whereas high consumption of sugar and saturated lipids accelerates telomere attrition. Those effects are likely to be globally mediated by oxidative stress and inflammation, as antioxidant and anti-inflammatory properties of nutrients are associated with longer telomeres. Physical activity may protect telomeres but more research is needed to establish a consensus on the optimal exercise dose (Figure 2). The beneficial effects of physical activity on telomeres could be driven by an increase in telomerase activity following an acute bout of exercise in PBMC, an alleviation of oxidative stress and a TERRA renewal in skeletal muscle. Further investigations are needed to study the other possible mechanisms contributing to the exercise-induced beneficial effects on telomere length and integrity.

We propose that engaging in a healthy diet and regular physical activity could be both promising strategies to protect telomere maintenance and improve health span at old age. However, more research on the molecular based mechanisms is required.

## Figures and Tables

**Figure 1 nutrients-10-01942-f001:**
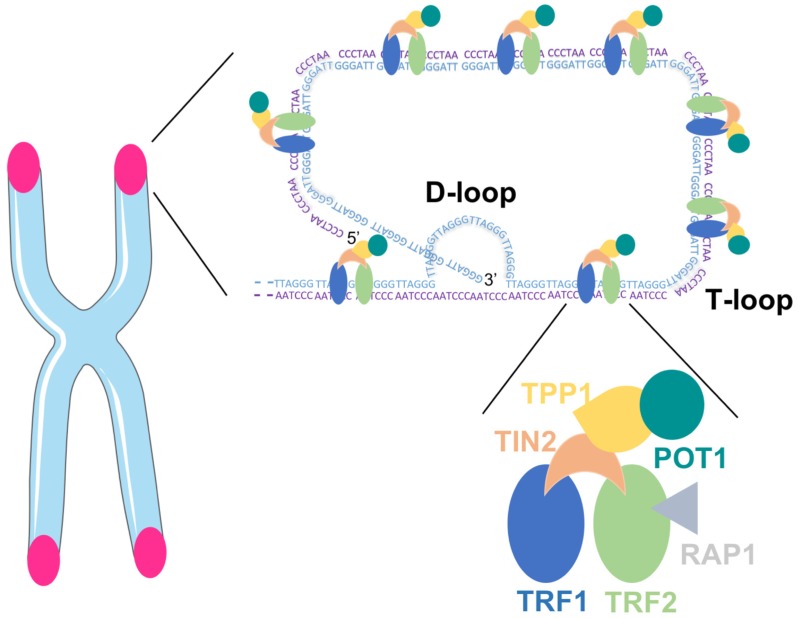
Telomeric DNA with the Shelterin complex facilitating the formation of D- and T-loop.

**Figure 2 nutrients-10-01942-f002:**
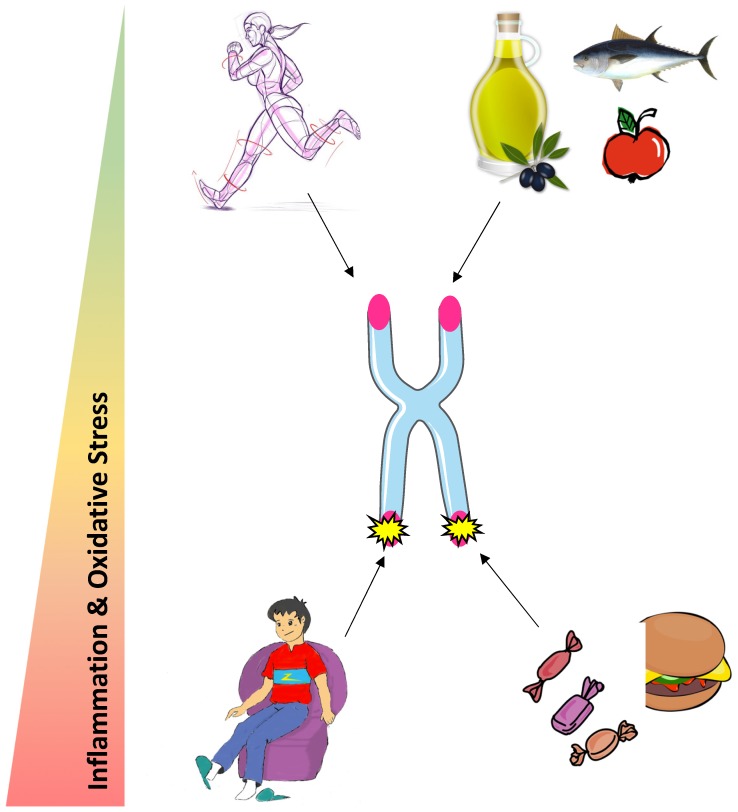
Potential influence of physical activity and nutrition on telomere health.

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
