# Peer review of "Physical Activity and Nutrition: Two Promising Strategies for Telomere Maintenance?"

_nutrients, 2018, doi:10.3390/nu10121942_

Round 1

Reviewer 1 Report

In this review, Balan et al., summarize the recent literature on the effects of nutrition and physical activity on telomere dynamics and aging. The review is well written and easy to read, a nice overview of what has been published recently on the impact of lifestyle on telomere maintenance. In some instances, the authors also indicated where there exist conflicts in the literature and have attempted to provide explanations as to why these may exist.

The only minor comments are the following:

I would suggest to invert the two sentences from line 67 to line 70 by writing first the sentence starting with “critical loss of telomeric DNA” and then “damage at telomeres”. This would make it easier to the readers to follow why telomere shortening induces DNA damage at telomeres.

Since ROS production is indicated in paragraphs 3.3, 4.3 and 4.4 and linked to telomere maintenance, it would be interesting to the readers to know that telomeric DNA is particularly sensitive to ROS as in vitro experiments have shown that telomeric DNA is more reactive to ROS than nontelomeric sequences (Oikawa and Kawanishi, FEBS Letter of 1999) and this may contribute to telomere shortening. The impact of ROS on telomere maintenance was also recently reviewed by Ahmed and Lingner, in Differentiation 2018. 

Author Response

Response to Reviewer 1 Comments

We thank the reviewer for the suggestions that helped us to improve the manuscript.

Point 1: I would suggest to invert the two sentences from line 67 to line 70 by writing first the sentence starting with “critical loss of telomeric DNA” and then “damage at telomeres”. This would make it easier to the readers to follow why telomere shortening induces DNA damage at telomeres.

Response 1: The two sentences have been inverted à "Critical loss of telomeric DNA or unprotected telomeres leads to insufficient chromosomes end protection and to the activation of the DNA damage response [17]. Damage at telomeres can also happen independently of cell division, notably in response to the accumulation of oxidative lesions, smoking behavior or obesity [22]."

Point 2: Since ROS production is indicated in paragraphs 3.3, 4.3 and 4.4 and linked to telomere maintenance, it would be interesting to the readers to know that telomeric DNA is particularly sensitive to ROS as in vitro experiments have shown that telomeric DNA is more reactive to ROS than nontelomeric sequences (Oikawa and Kawanishi, FEBS Letter of 1999) and this may contribute to telomere shortening. The impact of ROS on telomere maintenance was also recently reviewed by Ahmed and Lingner, in Differentiation 2018. 

Response 2: A small paragraph concerning the impact of ROS on telomeric DNA has been added at the beginning of section 3.3 as well as the references suggested by the reviewer.

"Unhealthy dietary habits have been linked to an inflammatory state, contributing to progressive telomere attrition [47]. As unhealthy dietary habits increase the production of reactive oxygen species (ROS), it is possible that the impact on telomere erosion goes through an increased oxidation of telomeric DNA. Supporting this, is the observation that, because of their high content in guanine residues, telomeric sequences are highly prone to oxidation into 8-oxoG, at least in in vitro experiments [48]. When present at telomeres, 8-oxoG residues are likely decreasing the affinity of shelterin proteins for telomeric DNA and are, as well, disrupting the G-quadruplex structures of telomeres that play important roles at telomeres, like the regulation of telomerase activity [49]. Altogether, it is therefore possible that nutrients regulate telomere health by regulating oxidative stress and systemic inflammation [50]."

Reviewer 2 Report

Studies about the relation between lifestyle factors (such as foods, diets and physical activity) and telomere maintenance are a challenging field of research. This review summarizes and discusses articles about this complex topic with some evidence for a relation based on selected data from clinical and pre-clinical basic research. This is a generally well written review with some shortcomings.

Specific concerns:

1. The topic has been already covered by other recent reviews not mentioned, such as J Gerontol A Biol Sci Med Sci. 2017,73(1):39; Int J Mol Sci. 2017, 18(12): 2573; Oncotarget. 2017,8(27):45008.

2. 50% of key studies showing no association between physical activity and telomere length as listed in one of these recent reviews (Int J Mol Sci. 2017; 18(12): 2573). Authors should mention/discuss this point accordingly.

3. Some minor points that should be corrected accordingly:

Line 50: telomeres consist of repetitive DNA G- and C- rich sequences.

Lines 86,87: adequate reference for the relation between telomere length and consumption of coffee or alcohol is missing.

Lines 173,174: inadequate reference

Line 179: inadequate references; study of mouse model (ref #47) should be mentioned

Lines 190-193:  sentence not clear

Line 207: inadequate reference

Line 250: somewhat inadequate reference, this cited review is about shelterin interactions and interplay with telomere functions and not about telomerase activity or TERT.

Line 267: incomplete information, TERF2IP alias RAP1 has extra-telomeric roles such as subtelomeric silencing, transcriptional regulation and NF-kappaB signaling as outlined in Nat Rev Cancer. 2011;11(3):161. At least the last topic is related to the mechanism described as inflammation in chapter 3.3 or the redox-sensitive transcription factor described in chapter 4.3 and should be mentioned.

Line 276: inadequate reference

Author Response

Response to Reviewer 2 Comments

We thank the reviewer for the suggestions that helped us to improve the manuscript.

Point 1 : The topic has been already covered by other recent reviews not mentioned, such as J Gerontol A Biol Sci Med Sci. 2017,73(1):39; Int J Mol Sci. 2017, 18(12): 2573; Oncotarget. 2017,8(27):45008.

Response 1 : The purpose of the present review was to give a larger overview of the effect of lifestyle (nutrition + exercise treated in the same manuscript) on telomere dynamics and not only telomere length. We have now added this rationale to the manuscript: " While previous reviews specifically focused on the regulation of telomere length by either nutrition [12] or exercise [13, 14], the present review will give a broader view looking at the impact of lifestyle variables on human telomere dynamics with emphasis on diet and physical activity."

Point 2 : 50% of key studies showing no association between physical activity and telomere length as listed in one of these recent reviews (Int J Mol Sci. 2017; 18(12): 2573). Authors should mention/discuss this point accordingly.

Response 2 : We thank the reviewer for this suggestion that we have now added at the beginning of section 4. "This section will deliberately present a positive view regarding the effects of physical activity on telomere dynamics, but it should be kept in mind that about half of the studies dealing with that topic found no association between physical activity and telomere length [13]. Obviously further investigation will be needed to determine why the different findings are such discrepant from one study to the other. In addition, new analytical tools need to be developed to measure telomere length more accurately as well as new biomarkers for assessing biological aging [13]."

Point 3 : Line 50: telomeres consist of repetitive DNA G- and C- rich sequences.

Response 3 : This comment has been corrected at line 50.Mammalian telomeres consist of repetitive DNA G- and C-rich sequences (5’-TTAGGG-3’/3’-CCCTAA-5’) with the 3’ end of the G-strand extending beyond the 5’ end [15].”

Point 4 : Lines 86,87: adequate reference for the relation between telomere length and consumption of coffee or alcohol is missing.

Response 4 : We added the reference of Liu et al. (2016) for the positive influence of coffee consumption on telomeres: “Coffee Consumption Is Positively Associated with Longer Leukocyte Telomere Lengh in the Nurses Health Study”. We also added the reference of Pavanello et al. (2011) for the negative impact of alcohol on telomere length: “Shortened telomeres in individuals with abuse in alcohol consumption”. "Telomere length is positively associated with the consumption of legumes, nuts, seaweed, fruits and 100% fruit juice, dairy products and coffee whereas it is inversely associated with consumption of alcohol, red meat or processed meat [27, 28, 33, 34]."

Point 5 : Lines 173,174: inadequate reference

Response 5 : We added the reference of Garcia-Calzon et al. “Dietary inflammation index and telomere length in subjects with a high cardiovascular disease risk from the PREDIMED-NAVARRA study : cross-sectional and longitudinal analyses over 5y” and adapted the sentence : “Unhealthy dietary habits have been linked to an inflammatory state, contributing to progressive telomere attrition [47]”.

Point 6 : Line 179: inadequate references; study of mouse model (ref #47) should be mentioned

Response 6 : We only kept the reference of the study on the mouse model. We can now read: "The intake of nutrients having antioxidant and anti-inflammatory properties such as vitamin C or E, polyphenols, curcumin or omega-3 fatty acid has been associated with longer telomeres, at least in mouse [51]."

Point 7 : Lines 190-193:  sentence not clear

Response 7 : We have now re-formulated the sentence as follows: "In 2015, a direct association was found between the pro-inflammatory capacity of the diet and telomere shortening in a population at high risk of cardiovascular disease. The diets with the higher pro-inflammatory scores were associated with a higher risk of having shorter telomeres and a two-fold risk of accelerated telomere shortening after a five-year follow-up period [47]."

Point 8 : Line 207 : inadequate reference

Response 8 : We apologize. This was a mistake. The reference has been corrected by the reference of Gomez-Delgado et al. (2014) “Polymorphism at the TNF-alpha gene interacts with Mediterranean diet to influence triglyceride metabolism and inflammation status in metabolic syndrome patients: From the CORDIOPREV clinical trial”.

Point 9 : Line 250: somewhat inadequate reference, this cited review is about the Shelterin interactions and interplay with telomere functions and not about telomerase activity or TERT.

Response 9 : We removed the sentence and the reference : “An increase in telomerase activity after exercise could be one mechanism”. We can now read : “ While physical activity has been associated with longer telomere length and protection against age-related telomere attrition [65,67-72], the mechanisms by which physical activity exerts its positive effects on telomeres are still largely unknown. As TERT, the catalytic subunit of the telomerase complex, is considered as the limiting factor for telomerase activity in human somatic cells, an increase in telomerase activity after exercise could promote telomere elongation.”

Point 10 : Line 267: incomplete information, TERF2IP alias RAP1 has extra-telomeric roles such as subtelomeric silencing, transcriptional regulation and NF-kappaB signaling as outlined in Nat Rev Cancer. 2011;11(3):161. At least the last topic is related to the mechanism described as inflammation in chapter 3.3 or the redox-sensitive transcription factor described in chapter 4.3 and should be mentioned.

Response 10 : This information has been added and we can now read: "However, TERF2IP/RAP1 has been found to be both a negative [76] and a positive regulator of telomere length [74]. Interestingly, TERF2IP/RAP1 is also known to play additional telomere-unrelated functions through the binding to extra-telomeric sites in the genome. Several regulatory functions have been attributed to the binding of TERF2IP/RAP1 outside telomeres, including the modulation of the nuclear factor-kappa B (NF-kB)-dependent pathway [77]. Whether the non-telomeric functions of TERF2IP/RAP1 play any role after exercise however warrants further investigation."

We removed the sentence : "The study of Chilton et al. showed that one intense endurance exercise bout was sufficient to regulate telomeric-associated genes, including TERT, and selective miRNA in white blood cells [73]."

Point 11 : Line 276: inadequate reference

Response 11 : We apologize. This was a mistake. The reference has been corrected by the reference of Cluckey et al. (2017) “Preliminary evidence that age and sex affect exercise-induced hTERT expression”.
